# Post-Migration Life Adversity and Mental Health of Refugees and Asylum Seekers: The Mediating Role of Resilience between Perceived Discrimination, Socio-Economic Strains, Structural Strains, and Mental Health

**DOI:** 10.3390/bs12070208

**Published:** 2022-06-24

**Authors:** Israel Fisseha Feyissa, Yeop Noh, Myeong Sook Yoon

**Affiliations:** 1School of Global Studies, Global Migration Research Canter, Kyungsung University, Busan 48434, Korea; israel12@jbnu.ac.kr or; 2Department of Social Welfare, Jeonbuk National University, Jeonju 54896, Korea; noh0824@jbnu.ac.kr

**Keywords:** asylum seekers, refugees, mental health, post-migration life adversity, resilience, perceived discrimination, socio-economic strain, structural strain, anxiety, depression

## Abstract

This study made a claim that perceived discrimination, socio-economic strain, and structural strain on displaced people have an adverse impact on their mental health. Our claim also acknowledges that these people potentially have a unique set of strengths and abilities that they rely on to overcome their immediate and future problems. The aim of this study is thus to examine the relationship between post-migration life adversity and mental health problems, and assess the potential mediating role of resilience among asylum seekers and refugees (219 asylum seekers and 42 recognized refugees) living in South Korea. Structural equation modelling was used to examine hypothesized pathways between post-migration life adversity, mental health and resilience. Fit indices showed adequate to excellent fit of the examined models with mental health as the outcome. Mental health was positively regressed on PMLA and negatively regressed on R. In addition, R partially mediated the association between PMLA and MH. In addition to providing the academic contributions of this study to the ongoing study of resilience and its social welfare implications, the result of the study indicated the necessity of improving the present and future socio-environmental factors that foster resilience among refugees and asylum seekers.

## 1. Introduction

Among the migrant groups, refugees and asylum seekers are the most vulnerable. Evidence suggests pre-migration suffering coupled with post-migration life adversities negatively affects mental health of these particular groups [1,2,3,4,5,6,7]. Symptoms of common mental disorders like depression, anxiety, and PTSD are much more common in asylum seekers and refugees as compared to the general population and other types of migrants [8]. Although they might not necessarily signify mental health problems, asylum seekers and refugees are victims of stressors that make day to day life less bearable [9,10].

Pre-migration factors, personal characteristics (e.g., age and gender), and post-migration experiences including acculturation challenges and receiving society receptivity is usually used to encompass the mental health challenges of refugees and asylum seekers [5]. The post-migration experience for the most part is similar between refugees and asylum seekers and their mental health status is often similar. However, legally unrecognized asylum seekers appear more vulnerable due to their unresolved legal refugee status [11]. Even after controlling for pre-migratory trauma predictors, it is reported that the levels of mental health problems are higher in asylum seekers compared to refugees with formal refugee status [12,13].

It is often apparent that recognized refugees and asylum seekers differ in many of their socio-economic aspects. For instance, the residence permit gives the recognized refugees a relative opportunity to resettle in the country that would allow them to have a job and build a social network. In contrast, asylum seekers are uncertain about obtaining a residence permit, and, therefore, their whole future is uncertain. Asylum seekers have just arrived in the country and most probably have a recent history of traumatic experiences. Therefore, the prolongation of time from arriving in a host country to the time of acquiring the full social protection could thus determine the mental health of the asylum seeker [14,15,16].

In the process of migration, the migrant is subjected to stress. In the first stage of migration, a stage of deciding and preparing to move, the individual might face relatively lower mental distress, whereas in the second stage (physical relocation) and the third stage of post-migration, mental stress will build up and reach a critical point. Especially the third stage of migration will have its toll as the migrant will be expected to learn new socio-cultural rules and assume a new role. For a refugees and asylum seekers, who are involuntarily migrating, the stress will escalate much faster as these people have less or no preparation to migrate to the new location. Post-migration environments were studied as having an impact on the mental health of asylum seekers after displacement [17,18,19,20,21,22,23]. Prolonged detention, insecure residency status, challenging refugee determination procedures, restricted access to services, and lack of opportunities to work or study, are studied as exacerbating symptoms of PTSD and depression [18,24,25,26,27,28].

Alongside the stress of the application process, asylum seekers also experience post-migratory social and financial hardship that adversely affects their mental health [26,29]. Social hardship here refers to social exclusion as well as the loss of social status and networks, and perceived discrimination in the host society. Financial hardship is associated to the restrictions placed by host governments on employment, livelihood, and social protection that result in the deprivation of basic needs and necessities. However, there are instances where recognized refugees also face the same social and financial hardships in the same way as asylum seekers.

The reoccurring daily stressors faced in the host country have been reported to be associated with the severity of common mental health symptoms among refugees and asylum seekers [30]. When compared to pre-migratory traumatic events, daily stressors have an equal or even stronger relationship with poor mental health [31,32]. Post-migration stressors such as discrimination by host society and social isolation contribute more to depression and anxiety than pre-migration traumatic experience [33].

In the population of refugees and asylum seekers, up-to-date information on the rate and characteristics of mental health conditions are still in great need. Before the 1970s, the field of study concerning mental health of refugees lacked scientific findings for an epidemiological understanding of prevalence and nature of mental health problems among refugees and asylum seekers. The addition of post-traumatic stress disorder (PTSD) in the Diagnostic Statistical Manual of Mental Disorders (DSM-III) set the foundation for the modern era of research in the refugee field [34]. However, a limited focus on pre-migratory trauma as the main determinant of mental health may have contributed to concealing the impact of concurrent social stressors found in refugee and asylum seekers living conditions [35,36].

When it comes to prior studies on mental health of refugees and asylum seekers living in South Korea, most or almost all generally discussed the overall settlement circumstances and related mental health issues of North Korean defectors to South Korea [37,38,39,40]. Unlike European states such as France or Germany, Korea’s refugee policy is not designed to separately treat refugees and asylum applicants; it is rather a clause of the immigration law [41]. The immigration law ‘require applicants for asylum to register with the immigration office, effectively revealing their illegal presence in Korea to the government agency responsible for control of illegal migrants’ [40]. This arrangement put refugees and asylum seekers under a constant legal scrutiny, rather than having a humanitarian approach that refugee laws usually have. The strict legal and procedural scrutiny is also reported to ‘push asylum applicants into the illegal migrant category’ which is often associated with a precarious life [41]. On a societal level, the willingness of the host society in accepting outsiders greatly affects the livelihood of the refugees and the asylum seekers. Although the immigration policy is always mentioned to determine the fate of these type of migrants, in the South Korean setting, the number of immigrants by visa status has always been determined by demand in the host country’s labor and marriage markets and by Korean citizens’ attitude towards immigration [42]. As a contextual background, the initial argument of this study for possible factors for post-migration life adversities among refugees and asylum seekers emanates from two ideal blocks. First, the post-migration experience of refugees and asylum seekers in South Korea, although different between the groups, is highly affected by the immigration policy, South Korean societal dynamics, and the overall refugee hosting mechanism (refugee status determination process and settlement programs). Second, due to such post-migration experience, mental health of refugees and asylum seekers is thus deemed to be shaped to a large extent by the mentioned social, economic, and environmental factors.

In as much as refugees and asylum seekers are facing life difficulties along with their exile, they are also among the resilient groups that push through life adversities. While definitions of resilience differ according to various contexts, it is often associated with a person’s ability to bounce back “following adversity and challenge and connotes inner strength, competence, optimism, flexibility and the ability to cope effectively when faced with adversity” [43]. Among the types of different migrants, refugees and asylum seekers are known for their resilience and their willingness to face a multitude of challenges to find safety and build a future, for themselves and their families. Although many refugees experience and witness traumatic events, the majority of those who survive do not develop significant mental disorders [44]. Refugees show enormous “courage and strength by coping with conditions of extreme deprivation and surviving against adversity” [45]. Once refugee status is established, a person has to deal with the demands of resettlement in a foreign country and loss and separation from their family and culture [21]. Despite this turmoil, researchers suggest many refugees go on to thrive in their new country and surroundings [46]. Professionals, however, continue to utilize a western medical model that places refugee experiences of hardship, deprivation, and distress in the terrain of psychopathology, rather than seeing it as a ‘normal’ response to an abnormal situation. As a result, refugee resilience is often obscured by the pervasiveness of the trauma narrative in refugees’ lives [47].

This study set out to investigate the post-migration life adversity and mental health of refugees and asylum seekers who are residing in South Korea. The study specifically focuses on the mediating role of resilience between perceived discrimination, socio-economic strains, structural strains, and mental health problems. Specifically, the purpose of this study is to explore whether difficult life adversities assert an influence on common mental health problems among refugees and asylum seekers living in South Korea, and further, to assess the potential mediating role of resilience.

## 2. Materials and Methods

### 2.1. Population Sample and Sampling Procedure

This study used a non-experimental, cross-sectional, and ex post facto survey research design. The target population for this study consisted of 42 refugees and 219 asylum seekers who are living in South Korea. The eligibility criteria for the study are determined after assessing the participant immigration status within the country and their duration of stay. Recruiting recognized refugees, asylum seekers, and the humanitarian status holders for the study was facilitated by the principal investigator of the study. The investigator contacted local NGOs and foreign communities to locate potential participants for the study. Local refugees and asylum seeker communities, which are located in Seoul, were the primary contact points to recruit potential participants. These communities are organized and are working on legal and settlement assistance for refugees and asylum seekers within South Korea. This research employed the convenience sampling technique in collecting data.

### 2.2. Data Collection Procedure

Jeonbuk National University’s institutional review board (IRB) approved the data collection. For accessibility convenience, the questionnaires were primarily converted into a Google form format with the necessary confidentiality precautions. The questionnaires were also translated into Amharic, Bengali, Burmese, Arabic, French, Swahili, and Vietnamese. Back translation was also employed for the first translated questionnaires. The responses to the questionnaires were directly recorded in the email address of the primary investigator. For participants who were unable to access the Google form, printed questionnaires were distributed.

### 2.3. Variables and Measurements

#### 2.3.1. Post-Migration Life Adversity

To measure post-migration life adversity, our study adopts a three sub-scaled measure from the Refugee Post-Migration Stress Scale (RPMS) [30]. Based on the definition of the post-migration life adversity of refugees and asylum seekers, the three adopted and modified subscales are: (1) perceived discrimination; (2) socio-economic strain; and (3) structural strain while living in South Korea as a refugee or asylum seeker. The measurement is a concise, multidimensional self-assessment instrument for the evaluation of post-migration life adversity among refugees and asylum seekers [48].

Each subscale comprises of three to four items, all scored on a 5-point Likert scale ranging from 1 (Never) to 5 (Very often). A sample item for the “perceived discrimination” subscale was: Please indicate how frequently you feel disrespected due to your nationality, refugee status, or asylum status by South Korean authorities.

#### 2.3.2. Mental Health Problems

To measure the common mental health problems of anxiety and depression, The Hopkins Symptom Checklist (HSCL-25) was used [49]. The HSCL-25 scale consists of 25 items; 10 items for anxiety symptoms and 15 items for depression symptoms. For all 25 items, the response is rated on a 4-point Likert scale ranging from 1 (Not at all) to 4 (Extremely). A higher score indicates more symptom severity. The questions on the measurement ask the participants to indicate how frequently they experience the stated mental health status. The scale has been validated in refugee populations [50,51].

#### 2.3.3. Resilience

The view of resilience in this study highlights the individual’s ability to effectively cope with significant adverse and threatening situations. The deduction of resilience in this study focuses on the individuals’ ability to “bounce back” to a previous state of normal functioning or using personal strengths and behaviors to avoid negative effects of adversities [52]. In addition, resilience is not a trait, but an ongoing continuous process of using interpersonal and intrapersonal capacities to successfully adapt to life stressors.

To measure resilience, the study adopted the Acculturation and Resilience Scale (AARS) [53]. This particular scale addressed a major gap in previous literatures about how to measure the positive acculturation and resilience of the newly arrived and relocated individuals. The scale’s psychometric properties designed for migrants and refugees from a range of ethnic communities are considered appropriate for our study to adopt it [53]. The AARS is originally a 44-item scale designed to measure acculturation and resilience of culturally and Linguistically Diverse Populations in Australia. The scale has item subsections for resilience, acculturation, and spirituality. This study only extracted 14 items for resilience, because the questions on the items encompass the initial conceptualization of resilience in this study. For each item, participants responded: 1 (do not agree), 2 (agree sometimes), 3 (agree mostly), or 4 (always agree).

The resilience variable needs to be understood to encompass the properties of an individual level resilience indicator and socio-ecological level indicator either at the same time or separately. For instance, the item ‘I can find many ways to solve a problem’ could be understood as an individual’s resourceful character trait that created the ability to solve ones’ problem. On the other hand, this particular item should also be understood as the socio-ecological environment allowing conditions for the individual to access solutions to his problems. The same analogy is also assumed for the rest of the resilience items.

#### 2.3.4. Socio-Demographic Variables

Data on gender, age, country of origin and year of immigration were collected. The data on asylum applicants and already recognized refugees were separated into a different group to investigate difference within groups. Age of respondents was categorized into two groups, as 18–40 years (early adults) and 41 and above (middle to later adults).

### 2.4. Statistical Analysis

The investigation of the thesis employed a method that could better account for the intricate circumstances of asylum seekers and refugees in South Korea. Given the challenges associated with acquiring substantive longitudinal data among these groups, mainly due to uncertainties surrounding residency and living conditions, a structural equation modeling (SEM) framework is considered appropriate to evaluate the implied causal assumptions. Although SEM cannot establish causality on the basis of observational data, it provides an opportunity to examine the plausibility of statistically modeled effects among variables drawn from theory and previous research [54]. Thus, this study aims to put the effects of post-migration life adversities on mental health of refugees and asylum seekers and the mediating effect of resilience in a statistically rendered structural model.

Initially descriptive statistics were conducted and structural equation modelling (SEM) was followed. The SEM adopted the two-step modelling approach [55]. The first step in this modeling is establishing the fit of the measurement model and the second step is establishing the fit of a model including the relationships between the latent variables, i.e., the full structural model. The statistical analysis employed mainly the Analysis of Moment Structures (AMOS) version 23.0 to test the causal relationships among the variables. The analysis process was also rechecked thorough Mplus V8.3 software (version 8, Linda Muthén and Bengt Muthén, 3463 Stoner Avenue, Los Angeles, CA 90066).

Confirmatory factor analysis (CFA) was also employed in order to examine the relationships among the different constructs within the conceptual model. To assess the measurement model in CFA, the analysis first considered the measurement model fit and then evaluated the validity of the measurement model. In CFA, indicating the difference between the endogenous and exogenous constructs is not needed. However, the structural model testing requires the distinction between the between endogenous and exogenous constructs. Measured variables are represented in rectangular shapes, while the covariance is usually represented by two-headed arrows, whereas a causal relationship from a construct to an indicator is represented by a one-headed arrow. In accordance with the analysis approach within SEM, all the measured items were treated as latent variables with the individual item responses modeled as reflective indicators. Post-migration life adversity was first modeled as three correlated first-order latent variables using the subscales of the RPMS perceived discrimination (four items), socio-economic strain (four items), and structural strain (four items). Resilience was modeled with 14 items as reflective of personal resilience indicators as it is adopted from the Acculturation and Resilience Scale (AARS). Mental health problems were modeled as a latent variable with the separate scores of anxiety and depression driven by the standardized scoring procedures of HSCL25. However, preliminary model identification for mental health problems was not rendered due to the variable being modeled with only two indicators. Thus, mental health problems were examined within the full structural model.

The recommended fit indices that should be considered in order to assess the model goodness-of-fit were also observed [56]. In this study, the goodness-of-fit indices used to evaluate the model fit are presented in Table 1.

## 3. Results

### 3.1. Participant’s Description

The target sample for this study was refugees and asylum seekers who are living in South Korea. These individuals have a post-migration refugee experience either as an asylum seeker or as a recognized refugee within South Korea. A total of 350 (*n* = 250 asylum seekers, *n* = 100) survey invitations were sent out to potential participants. The eligible participants were selected by checking their immigrant status for the immigrant group and by checking national registration Korean status for Korean participants. A minimum of a 6-month stay in South Korea was a mandatory participant selection criterion. Screening questions for a past history of mental health problems before migrating to South Korea before this particular survey were asked and those with a history mental health were excluded from the survey. A total of 261 responses (*n* = 219 asylum seekers, *n* = 42) were retained after screening for missing data. Within the 219 asylum seekers’ responses, 20 responses were collected from humanitarian status holders.

As it stated in Table 2, the majority of the participants are male, making up 80.8% of the asylum seekers and 64.2% of the recognized refugees. The majority of the respondents are also above the age of 31 (55.8% of the asylum seekers and 59.6% of the recognized refugees). The highest level of education attained by the majority of the asylum seekers is distributed between pre-high school and college graduate. Within the sample of recognized refugees, the highest level of education attained is distributed between pre-high school and college graduate. The majority of the asylum seekers stayed in South Korea for a minimum of 1 year and a maximum of 5 years. On the other hand, the majority of the recognized refugees stayed in Korea for a minimum of 3 years and a maximum of more than 10 years. The origins of the participants are from Africa, the Middle East, and Southeast Asian countries.

### 3.2. Preliminary Data Analysis

The preliminary examination of the data is intended to detect missing data, outliers, as well as normality and homogeneity of the sample data file. The missing data within the study sample is found to be distributed in a random manner and were below 2%. To replace missing data for the categorical variables, the mean substitution method is used. However, missing data for nominal variables were excluded during the multi-group analysis.

The post-migration life adversity construct scored a Cronbach alpha score of 0.936. The mental health construct and the resilience construct scored Cronbach alpha scores of 0.976 and 0.703, respectively.

### 3.3. Descriptive Statistics of Measurements

#### 3.3.1. Post-Migration Life Adversity

As displayed in Table 3, the post-migration life adversity measurement was measured using a 5-point Likert scale. The total mean for PMLA measurement ranges between 3.82 (±1.380) and 4.59 (±0.931) indicating a severe level of adverse post-migration experience. Between groups, the means for the items related to PMLA range between 3.90 (±1.28) and 4.59 (±0.875) for the asylum seekers, while the means for the items related to PMLA range between 3.9 (±1.55) and 4.79 (±0.782) for the recognized refugees.

#### 3.3.2. Mental Health

The mental health measurement HSCL-25 was measured using a 4-point Likert scale. The total mean for the measurement for anxiety ranges between 2.51 (±1.30) and 3.69 (±0.733) while the total mean for the measurement of depression ranges between 2.66 (±1.28) and 3.47 (±0.947).

Between groups, the means for the items related to anxiety range between 2.44 (±1.29) and 3.68 (±0.740) for the asylum seekers and 2.86 (±1.20) and 3.71 (±0.70) for the recognized refugees. The results indicated that the asylum seekers and the recognized refugees have experienced a severe to extremely severe level of anxiety while seeking asylum or even as recognized refugees. The means the items related to depression range between 2.51 (±1.26) and 3.38 (±0.00) for the asylum seekers and 3.14 (±1.26) and 3.79 (±0.56) for the recognized refugees. The results for depression also indicated that the asylum seekers and recognized refugees have experienced a severe to extremely severe level of depression while seeking asylum or even as recognized refugees.

#### 3.3.3. Resilience

Resilience using AARS was measured using a 4-point Likert scale. The results in Table 3 show the total mean for resilience ranges between 1.23 (±5.41) and 3.79 (±0.48). Between groups, the means for the items related resilience range between 1.26 (±0.57) and 3.82 (±0.45) for the asylum seekers and 1.07 (±0.261) and 3.64 (±0.61) for the recognized refugees.

### 3.4. Analysis of Measurement Models

#### Goodness of Fit Indices

Given the sensitivity of χ^2^ statistics to sample size and model complexity, the application of several indices of fit increases the chances of accepting a well-fitted model and reduces the chances of declining a well-fitted model based on a single index [59].

For potential fit improvements, modification indices (MIs) were considered. By applying a Satorra–Bentler chi-square difference likelihood ratio (∆S-Bχ^2^) test, comparison between nested models was performed. A non-significant Satorra–Bentler chi-square implies a model with larger degrees of freedom does not worsen the model fit. Thus, maximum likelihood estimation with robust standard errors (MLR) was used for all analyses.

In the first modeling, the fit indices for the measurement models of resilience as measured by the Acculturation and Resilience Scale (AARS), there was an acceptable fit index with a significant Satorra–Bentler chi-square. However, with a re-specified model that allowed a free estimation of the error terms of four items, this resulted in significant enhancement of model fit indicated by a non-significant Satorra–Bentler chi-square. The re-specified model resulted in S-Bχ^2^ (11.79, df = 8, *p* = 0.338) with other acceptable model fit indices. The re-specified fit for the resilience measurement showed a significant improvement of the model fit, i.e., ∆S-Bχ^2^ = 25.72, df = 1, *p* < 0.001 (see Table 4).

In the first modeling, the fit indices for the measurement models of post-migration life adversity measured by 12 items, there was an acceptable fit index with a significant Satorra–Bentler chi-square. However, with a re-specified model that allowed a free estimation of the error terms of two items, this resulted in significant enhancement of model fit indicated by a non-significant Satorra–Bentler chi-square. The re-specified model resulted in S-Bχ^2^ (87.79, df = 32, *p* = 0.249) with other acceptable model fit indices. The re-specified fit showed a significant improvement of the model fit, i.e., ∆S-Bχ^2^ = 1.73, df = 1, *p* < 0.001 (see Table 4).

### 3.5. Full Structural Model

After maintaining the goodness-of-fit analysis to the measurement models, the hypothesized mental health problem was selected as the outcome variable. Then, the fit for the full structural model was assessed with mental health as outcome variable.

The model fit for the full structural equation model is displayed in Table 5. Although there was expected significant S-Bχ^2^ (S = 362.52, df = 175, *p* < 0.001) because of the model complexity caused by the large DF and sample size, the fit indices for the full structural model also resulted in satisfactory adequate fit to the data by RMSEA = 0.053 (90% CI = 0.045–0.051), CFI = 0.921, and SRMR = 0.052.

The replacement of mental health outcomes with scores of HCSL 25-anxiety and HCSL 25-depression was necessary at this stage. The replacement of the outcomes of mental health with separate scores of anxiety and depression and the ∆S-Bχ^2^ test for comparison with the full structural model showed less damage to the model fit (∆S-Bχ^2^ = 2.541, df = 3, *p* = 0.451).

Within the full structural equation model, post-migration life adversity significantly influenced mental health of refugees and asylum seekers. According to the result rendered on the full structural model, mental health was significantly regressed on post-migration life adversity (B = 1.687, 95% CI = 0.480–1.031). In other words, the higher score on post-migration life adversity is associated with the higher levels of the mentioned mental health states, i.e., depression and anxiety.

Post-migration life adversity also significantly and negatively influences resilience of refugees and asylum seekers. The finally rendered full structural model showed that resilience, in turn, was negatively regressed on post-migration life adversity. This result is an indication that higher levels of post-migration life adversity were associated with lower levels of resilience (B = −0.712, 95% CI = −0.885 to −0.357).

Resilience significantly and positively influencing mental health of refugees and asylum seekers is also supported in the full structural model. The relationship between mental health and resilience also indicated that the higher resilience score is associated with lower levels of depression and anxiety. Mental health was significantly and negatively regressed on resilience (B = −0.138, 95% CI = −0.144 to −0.032).

There is also a difference in effects of age on mental health and resilience between young adulthood (18–30) and early and late adulthood (31 and above). However, there was no significant difference of the effect of age on post-migration life adversity. Young adulthood was associated with higher levels of depression and anxiety (B = 0.519, 95% CI = 0.068–0.418). Young adulthood is also associated with high levels of resilience (B = 0.457, 95% CI = 0.48–0.212).

Being female showed a significant association with both mental health and post-migration life adversity. Being female was associated with higher levels of depression and anxiety (B = 0.789, 95% CI = 0.82–1.12) and higher levels of post-migration life adversity (B = 0.592, 95% CI = 0.72–0.92). However, there is no significant association between being female and resilience.

Being an asylum seeker or having a G1 visa status showed a significant association with mental health and resilience. Being an asylum seeker was associated with higher levels of depression and anxiety (B = 0.562, 95% CI = 0.098–1.02) and higher levels of resilience (B = 0.357, 95% CI = 0.76–1.22). However, there is no significant association between being an asylum seeker and post-migration life adversity.

The final and fully structured equation model with the significant associations and corresponding estimates can be viewed in Figure 1. In the full structural equation model, the estimated regression weights are unstandardized (B) with robust standard error in parentheses. The significant regression weights and the weight estimates of main interest are represented in solid-line arrows and bold. The non-significant regression weights are represented in dotted lines. The values for residual variance are indicated in arrows and they signify that variances were also part of the analysis of the full structural equation model.

The full structural equation model accounted for a substantive significant proportion of the variance in mental health problems. The proportion of variance in the mental health problems which can be predicted from the independent variables indicates a value of 58.9% (R^2^ = 0.589, SE = 0.052, *p* < 0.001). As can be referred from Table 6, within the mediation analysis a bias-corrected bootstrapped 95% confidence interval was applied. The CI was based on 1000 bootstraps and the bootstrapped CI shows that resilience partially mediated the association between post-migration life adversity and Mental Health Problems (indirect effect estimate = 0.871, 95% CI = 0.044–0.183).

## 4. Discussion

The finding rendered from the structural equation model confirmed the post-migration life experience of refugees and asylum seekers as an instigating factor of adverse mental health outcomes. Within the final full model, perceived discrimination, socio-economic strain, and structural strain showed a significant influence and showed a sufficient fit to the sampled data. This significant confirmation that post-migration life adversity negatively influences the mental health of refugees and asylum seekers asks for a broad understanding of extended negative factors that affect displaced people pre- to post-migration experience.

The important finding in this study is that resilience of refugees and asylum seekers is a mediating factor between post-migration life adversity and mental health problems. This finding highlights how the strength and adaptability skills of the refugees and the asylum seekers become instrumental in minimizing the damage along the pathway of negative post-migration life experience leading to poor mental health. Previous studies that focused on the role of resilience among displaced people like refugees and asylum seekers indicated resilient personal qualities like optimism, adaptability and perseverance helped them to cope and survive post-migration adversities [60]. Utilizing inner belief in one’s own inner strength to deal with life’s challenges, ref. [61] a positive attitude, and having hope for a good future helped refugee women to cope [62]. Previous studies associated resilience with the individual’s determination to take control of their life challenges, rather than being a victim [63]. Looking ahead to the future despite challenges, acceptance of the situation, and refocusing on the present and the future are also reported to strengthen refugees’ resilience [64,65]. Resilience as a mediator is also an indicator of how post-migration life adversity of refugees and asylum seekers may also indirectly impact mental health by hampering personal will and mental strength to go through adverse life events from the asylum-seeking process to settling as a recognized refugee.

One highlight of the result which is related to post-migration life adversity and immigration status indicated immigration status (acquiring a recognized refugee status) has no significant relationship with post-migration life adversity. Although immigration status has a significant effect on mental health problems and resilience, the non-significant relationship with post-migration life adversity is indicative of the similarity of post-migration life experience between the asylum seekers and recognized refugees. In other words, the attainment of the F-2 visa status would have produced a different life for the recognized refugees after recognition. However, the result of the structural equation model is indicative of no difference in perceived discrimination, socio-economic strains, and structural strains among the two groups. For the asylum seekers, psychosocial stressors related to the extended insecure status, perceived discrimination and restrictions during the asylum-seeking process may increase the post-migration life adversity. However, the non-significant result could help to question the outcomes of socio-cultural receptiveness of the country, the allocation within the immigration policy, and the settlement process after the refugee status acquisition.

On the same note, the results of the full structural equation model show that sex has a significant association with mental health problems and post-migration life adversity. In other words, being a female refugee or asylum seeker is potentially associated adverse post-migration life experiences. However, sex does not have a significant relationship with resilience, indicating the fact that individual qualities of resilience either as a male or female do not have a different effect within the rendered model. In this particular result it is evident that, for the female participants of this study, the post migration experience is evidently challenging enough to produce an adverse mental health status. However, when it comes to resilience varying between males and females, this result could also inform us that the type of qualities that enhanced individual resilience did not differ according to the gender type. In other words, how male and female respondents answered the indicators within the resilience measure have no significant difference.

When it comes to age, it has a significant association with mental health problems and resilience. However, age does not have a significant association with post-migration life adversity. This particular result is indicating the fact that the effect of perceived discrimination, socio-economic strain, and structural strain is not different between the age groups of 18–30 and 31 and above.

All in all, the result of the study could point to public health intervention approaches that focus on solutions that aim to restore the individual inner strength, the interactions that refugees and asylum seekers have with their surroundings, and a system that creates particular ‘opportunities for personal growth that are available and accessible’ [66]. Therefore, the results suggest the importance of a public health approach that address the mental health of refugees and asylum seekers on an individual level by weighting the ‘meaningfulness of these opportunities and the quality of the resources provided’ [66]. Such approach would thus require increased collaboration between immigration policy makers, mental health services, and refugee resettlement programs.

According to complex adaptive systems theory (CAS) the resilience of an individual over the course of development depends on the function of complex adaptive systems that are continually interacting and transforming [52]. Complex adaptive systems of resilience in the context of refugees and asylum seekers will look into types of risk factors within the post-migration experience, the type of individual adaptive behavior, and exogenous factors that potentially affect the individual post-migration experience. Based on CAS assumptions, the resilience of refugees and asylum seekers will be maintained only if the external factors in the setting of refugees and asylum seekers work together to achieve that. These external factors might include anything in the wider sphere of social, cultural, political, and economic environment factors that have a potential to affect the refugee or the asylum seeker. On the socio-cultural and political level, the willingness to accommodate displaced people like refugees and asylum seekers will contribute a lot to resilience. In other words, the complex intricate systems surrounding the refugee and the asylum seeker captures the intelligent trade off negotiations between capacity to cope and other types of variability in order to cope better to future stressors [67]. In simpler terms, the context of refugees and asylum seekers in South Korea needs the necessary tradeoff between the respective socio-cultural context of the host country, the affecting immigration policy, the status determination process, and the level of support to individual attributes that produces resilience. In the same line of argument, a particular result of this study about the immigration status (the status change from asylum seeker to a recognized refugee) having no significant relationship with post-migration life adversity also recommends that at least the recognition of the F-2 Visa as an exogenous factor should be able to help the refugee access better resources.

Individualistic views of resilience assume the individual responsibility for risk aversion [68], implying that the responsibility to display adaptive functioning and mental health is principally on the individual and not on the environment [69]. However, as Schiltz and colleagues’ [68] study questioned, it is also important to put in question how access to resources is granted or denied within refugee and asylum policy, the socio-ecological position of the individual, and resources at their disposal determine the resilience of refugees or asylum seekers. The socio-ecological view of resilience repositions resilience and successful transition processes in the social sphere and provides a second way to understand resilience beyond the individual sphere. In the socio-ecological understanding of resilience, resilience is understood in terms of person–environment interactions, involving multilevel dynamics in the larger social ecology [69]. When it comes to refugees and asylum seekers, the concept of resilience within policy and practice initiatives have ignored the importance of an enabling environment and how granting access to resources allows refugees and asylum seekers to become resilient. The socio-ecological model of resilience is focusing on how to revitalize the resourcefulness of refugees and asylum seekers [66]. Individual resourcefulness could appear as a character trait, but resourcefulness is how much the individual appears resourceful in the utilization of their social networks (social capital), their respective communities, or government support systems. It is proper to assume that because of the circumstances of the refugees and asylum seekers, the resources at hand are also similar for these groups. However, individual resourcefulness in utilizing resources provided within the host country might differ.

Germain’s [70] study mentioned how individuals in a social setting constantly strive to achieve person–environment fit (learning and adapting) under changing circumstances. Similar to the resilience ecological stress model, person-environment fit could be interpreted as refugees and asylum seekers dealing with or adapting to ‘daily stressors as well as to adverse critical events, while still maintaining effective functioning’. This ecological perspective of resilience demands suitable conditions for refugees and asylum seekers’ adaptation to be associated with how well they meet the demands of their surroundings.

## 5. Conclusions

In this particular study, it is established that the role of resilience mediates between post-migration life adversities and mental health. This result is indicative of resilience as a partially lessening factor for psychopathology after or within highly adverse events. In other words, perceived discrimination, the socio-economic strains, and the structural strains are ever present in the lives of the refugees and asylum seekers, yet their personal and socio-ecological resilient traits such as seeking out and using ample personal, material, and social resources promoted mental health.

The progress on the study of resilience also urged the need to constantly understand the contexts in which various types of resilience occur. Studies so far agree that ‘resilience is a complex construct that may have specific meaning for a particular individual, family, organization, society and culture; that individuals may be more resilient in some domains of their life than others, and during some phases of their life compared with other phases; and that there are likely numerous types of resilience that depend on context’ [71]. Although this study has a limitation of being a cross-sectional study, the measures for the variables had room for respondents to reflect on their migration experience in South Korea. Within the scope of investigation, at least within the respondent’s post-migration encounters, whether it is in their socio-cultural contacts or their hurdles within the refugee status determination, they displayed resilient traits that are enough to lessen the burden of mental health problems. By any means, however, the type and context of resilience mentioned in this particular study shouldn’t be duplicated for other settings of refugees and asylum seekers elsewhere other than the plausibility of resilience being a mediator between post-migration life adversities and mental health problems.

Current studies that focus on understanding the determinants of resilience agree on the promotion of protective factors that emanate from the individual to the surrounding environment. It is also important to recognize that successful determinants of resilience may vary from one person to the next based on multiple factors such as personality, specific challenges, available resources, and the environmental context. In this particular study, however, a significant factor that can boost resilience among the refugees and asylum seekers in South Korea is evidently found within equipping and eradicating challenges within their post-migration experience.

Within this study it is critical to understand that the subjects of this study are surrounded by families (although they might differ in degree from one another), their families in various communities, and these communities in societies and cultures. Thus, as recommended by previous studies, ‘interventions targeted at any one of these levels will impact functioning at other levels’ [71]. Sometimes the most effective strategy to enhance resilience at a specific level may involve intervening on a different level. For example, enhancing resilience in an asylum applicant may be more effective by equipping the immigration officers with resources to guide and support applicants along the way of the refugee recognition process than to intervene at the level of the asylum applicants. Similarly, the wider communities may enhance individual resilience and adaptation by providing the necessary resources for positive transformation. In other words, resilience in the individual is highly dependent on multiple layers of society [71].

## Figures and Tables

**Figure 1 behavsci-12-00208-f001:**
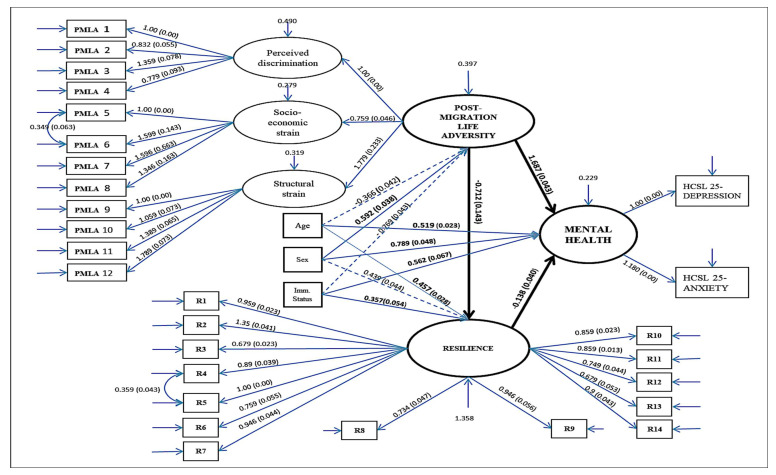
Full Structural Equation Model of Mental Health Problems (depression and anxiety) regressed on post-migration life adversity and resilience.

**Table 1 behavsci-12-00208-t001:** Criteria for goodness-of-fit.

Model Fit Index	Recommended Value	Reference
Satorra–Bentler scaled chi-square (S-Bχ^2^)	Non-significant at *p* < 0.05	[56]
Comparative fit index (CFI)	Cut-off values of ≥0.90 indicating adequate fit	[57]
Root mean square error of approximation (RMSEA)	90% confidence intervals with RMSEA < 0.06 indicating good fit	[57]
Standardized root mean square residual (SRMR)	SRMR < 0.08 indicating good fit	[58]

**Table 2 behavsci-12-00208-t002:** Participants’ description.

		Asylum Seekers (*n* = 219)		Recognized Refugees (*n* = 42)	
Item		N	%	N	%
**Sex**	Male	177	80.8%	27	64.2%
	Female	42	19.2%	15	35.8%
**Age**	18–30	97	44.2%	17	40.4%
	31 and above	122	55.8%	25	59.6%
**Level of Education**	Pre-highschool	82	37.4%	12	28.5%
	Highschool graduate	66	30.1%	9	21.5%
	College graduate	59	26.9%	21	50%
	Graduate school	12	5.4%	-	-
**Duration of stay in Korea**	Less than a year	6	2.7%	-	-
	>1 year < 3 years	81	36.9%	-	-
	>3 years <5 years	114	52%	15	35.7%
	>5 years <10 years	18	8.2%	12	28.5%
	More than 10 years	-	-	15	35.7%
**Origin**	Afghanistan	3	1.3%	-	-
	Angola	3	1.3%	-	-
	Bangladesh	8	3.7%	2	4.7%
	Burundi	3	1.3%	-	-
	Chad	6	2.7%	-	-
	Congo	6	2.7%	-	-
	Egypt	30	13.7%	6	14.2%
	Eritrean	4	1.9%	2	4.7%
	Ethiopian	38	17.3%	10	23.8%
	Ghana	6	2.7%	-	-
	Kenya	6	2.7%	-	-
	Liberia	9	4.1%	-	-
	Morocco	10	4.5%	4	9.5%
	Myanmar	7	3.1%	9	21.5%
	Nigeria	10	4.5%	2	4.7%
	South Africa	3	1.3%	-	-
	South Sudan	6	2.7%	-	-
	Syria	12	5.4%	2	4.7%
	Turkey	12	5.4%	1	2.3%
	Uganda	17	7.8%	1	2.3%
	Vietnam	10	4.5%	2	4.7%
	Yemen	10	4.5%	1	2.3%

**Table 3 behavsci-12-00208-t003:** Descriptive statistics of variables.

	Item Id.	Indicators	Total		Asylum Seekers (*n* = 219)		Recognized Refugees (*n* = 42)	
			M	SD	M	SD	M	SD
**Perceived discrimination**	**PMLA 1**	Discriminated by authorities	4.5	0.90	4.5	0.879	4.5	1.06
	**PMLA 2**	Feeling disrespected due to (my national background, refugee, or asylum status)	4.4	0.93	4.4	0.907	4.5	1.06
	**PMLA 3**	Racist encounters	4.4	0.97	4.3	0.993	4.7	0.805
	**PMLA 4**	Discrimination in the workplace	4.5	0.93	4.5	0.908	4.7	1.04
**Socio-economic strain**	**PMLA 5**	Being unable to access/buy necessities	3.8	1.3	3.7	1.33	3.9	1.59
	**PMLA 6**	Worry about the unstable financial situation	4.1	1.2	4.0	1.21	4.3	1.18
	**PMLA 7**	Unable to sustain oneself/family	3.9	1.2	3.9	1.19	3.9	1.55
	**PMLA 8**	Frustration due to loss of status	4.4	1.1	4.5	0.875	3.5	1.94
**Structural strain**	**PMLA 9**	Feeling excluded or isolated in Korean society	4.5	1.0	4.4	1.03	4.7	0.782
	**PMLA 10**	Unable to see improved future for self/family	4.2	1.0	4.2	0.982	4.1	1.42
	**PMLA 11**	Unable to access services	3.9	1.3	3.9	1.28	4.0	1.50
	**PMLA 12**	Unable to fully integrate into the society	4.5	0.8	4.5	0.814	4.7	0.805
**Anxiety measures**	**MH 1**	Suddenly scared for no reason	3.3	0.97	3.3	1.0	3.4	0.73
	**MH 2**	Feeling fearful	2.9	1.1	2.8	1.1	3.3	0.98
	**MH 3**	Faintness, dizziness, or weakness	2.6	1.2	2.5	1.2	3.0	1.1
	**MH 4**	Nervousness or shakiness inside	2.6	1.2	2.5	1.2	3.3	0.98
	**MH 5**	Heart pounding or racing	2.6	1.2	2.5	1.2	3.0	1.2
	**MH 6**	Trembling	2.5	1.3	2.4	1.3	2.8	1.2
	**MH 7**	Feeling tense or keyed up	2.5	1.2	2.4	1.2	3.0	1.1
	**MH 8**	Headaches	3.6	0.73	3.6	0.74	3.7	0.70
	**MH 9**	Spells of terror or panic	2.8	1.2	2.7	1.2	3.4	1.1
	**MH 10**	Feeling restless, cannot sit still	2.7	1.2	2.6	1.2	3.2	1.0
**Depression measures**	**MH 11**	Feeling low in energy-slowed down	2.8	1.2	2.7	1.2	3.5	0.91
	**MH 12**	Blaming yourself for things	2.9	1.1	2.8	1.2	3.4	0.91
	**MH 13**	Crying easily	3.2	1.0	3.1	1.1	3.7	0.59
	**MH 14**	Loss of sexual interest or pleasure	2.6	1.3	2.5	1.3	3.3	1.2
	**MH 15**	Poor appetite	2.6	1.3	2.5	1.3	3.1	1.2
	**MH 16**	Difficulties falling asleep, staying asleep	3.1	1.1	3.0	1.1	3.7	0.56
	**MH 17**	Feeling hopeless about the future	3.1	1.0	3.0	1.1	3.5	0.91
	**MH 18**	Feeling blue	2.6	1.2	2.5	1.2	3.4	1.1
	**MH 19**	Feeling lonely	3.4	0.94	3.3	1.0	3.9	0.26
	**MH 20**	Thoughts of ending your life	3.2	0.98	3.1	0.99	3.7	0.80
	**MH 21**	Feeling trapped or caught	3.3	0.99	3.2	1.0	3.7	0.56
	**MH 22**	Worrying too much about things	3.3	1.0	3.2	1.0	3.7	0.59
	**MH 23**	Feeling no interest in things	3.2	1.0	3.1	1.0	3.6	0.72
	**MH 24**	Feeling everything is an effort	3.1	1.0	3.1	1.0	3.5	0.73
	**MH 25**	Feelings of worthlessness	3.3	1.0	3.2	1.0	3.6	0.90
**Resilience measures**	**R1**	I can find many ways to solve a problem	3.79	0.484	3.82	0.450	3.64	0.618
	**R2**	I am able to cope with new situations	2.68	0.537	2.77	0.455	2.21	0.682
	**R3**	In a difficult situation, I usually find my way out	3.69	0.535	3.72	0.479	3.50	0.741
	**R4**	I am confident with my personal strengths	3.68	0.578	3.70	0.542	3.57	0.737
	**R5**	Although adapting is difficult, I am doing fine	3.67	0.601	3.71	0.562	3.43	0.737
	**R6**	I know where to get help when in trouble	1.60	0.852	1.63	0.901	1.43	0.501
	**R7**	I have a support system that can help me through the difficulties	1.31	0.768	1.31	0.759	1.36	0.821
	**R8**	I am open to learn new ways to communicate	3.61	0.595	3.63	0.587	3.50	0.634
	**R9**	I know about available public services	1.55	0.759	1.56	0.799	1.50	0.506
	**R10**	I can manage my two worlds	3.13	0.745	3.13	0.764	3.14	0.647
	**R11**	I have made close friends in South Korea	1.23	0.541	1.26	0.575	1.07	0.261
	**R12**	It does not worry me that I am from another cultural background	1.66	0.982	1.71	1.01	1.36	0.727
	**R13**	I support others who are in the same situation as me	1.64	0.845	1.56	0.795	2.07	0.973
	**R14**	I can take care of myself in a new place	3.33	0.708	3.38	0.716	3.00	0.562

**Table 4 behavsci-12-00208-t004:** Fit indices.

Fit Index	Models					
	Resilience (AARS)			Post-Migration Life Adversity (PMLA)		
	Model A: Uni-Dimensional	Model A: Uni-Dimensional	Model B vs. A	Model A: Uni-Dimensional	Model B: Re-Specified Uni-dimensional	Model B vs. A
*X* ^2^	48.52	96.25		96.25	87.79	
df	10	40		40	32	
*p*	<0.001	<0.001		<0.001	0.249	
CFI	0.935	0.977		0.977	0.852	
RMSEA (90% CI)	0.098 (0.061–0.13)	0.089 (0.041–0.11)		0.089 (0.041–0.11)	0.069 (0.27–0.072)	
SRMR	0.021	0.022		0.022	0.022	
∆S-Bχ^2^			25.72			1.73
∆df			1			1
*p*			<0.001			<0.001

**Table 5 behavsci-12-00208-t005:** Model Fit Indices for the full structural equation model.

Fit Index	Full Structural Equation Model
*X* ^2^	362.52
df	175
*p*	<0.001
CFI	0.921
RMSEA (90% CI)	0.053 (90% CI = 0.045–0.051)
SRMR	0.052
∆S-Bχ^2^	2.541
∆df	3
*P*	0.451

**Table 6 behavsci-12-00208-t006:** Direct, indirect, and total effect estimates.

		Unstandardized Estimate	BC 95% CI
**Direct effect**	Post-migration life adversity → Mental health problems	1.687	**0.480–1.031**
**Indirect effect**	Post-migration life adversity → Resilience→ Mental Health problems	0.871	**0.044–0.183**
**Total effect**	**Direct effect + Indirect effect**	**2.56**	**0.675–1.133**

## Data Availability

The data presented in this study are available on request from the corresponding author. The data are not publicly available due to ethical reasons.

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
