# Peer review of "Post-Migration Life Adversity and Mental Health of Refugees and Asylum Seekers: The Mediating Role of Resilience between Perceived Discrimination, Socio-Economic Strains, Structural Strains, and Mental Health"

_behavsci, 2022, doi:10.3390/bs12070208_

Round 1
Reviewer 1 Report
This is a very interesting article that brings fresh knowledge about the complex relationship between the condition of refugee and asylum seeker and the development of mental health problems. The quantitative method that is used in the study is clearly presented and sufficiently justified. The role of resilience as mediating factor between post migration life adversity and mental health problems is well developed. However, some surprising results, as the irrelevance of immigration status (asylum seekers recognised refugees),would have needed a more robust analysis, given the fact that it partly contradicts the CAS theory in respect to external factors. Other aspects as the relationship between sex and mental health (and resilience) (the gender factor ) would need further elaboration. Some qualitative elements concerning the context would enrich the conclusions.
Author Response
A response to reviewer 1
The research team for this paper wants to thank you for your interest and time to review our paper entitled ‘Post-migration life adversity and mental health of refugees and asylum seekers: The mediating role of resilience between perceived discrimination, socio-economic strains, structural strains, and mental health'. We appreciate all the comments given to the manuscript. They were highly insightful and very helpful in revising the study.
According to your suggestion, 'the irrelevance of immigration status (asylum seekers recognized refugees), would have needed a more robust analysis, given the fact that it partly contradicts the CAS theory in respect to external factors'. We also acknowledge that there is rather a weak argument in the discussion supporting this particular result. This particular part of the analysis is however highlighting the fact that the immigration statuses, being asylum seeker or a recognized refugee will have no significant difference in perceived discrimination, socio-economic strains, and structural strains. Normally attaining of the F-2 visa status (recognized status as a refugee) would have produced a different life for the recognized refugees after recognition; unfortunately that's not the case according to this result. Therefore, for clarity, we included an additional statement in the discussion part in line with the discussion about CAS as;
In the same line of argument, a particular result of this study about the immigration status (the status change from asylum seeker to a recognized refugee) having no significant relationship with post-migration life adversity also recommends, at least the recognition, F-2 Visa as exogenous factor, should be able to help the refugee access better resources.
You also asked us to include some elaboration on the relationship between sex and mental health (and resilience) (the gender factor). We added the following sentences to further explain the context of those results.
In this particular result it is evident that, for the female participants of this study the post migration experience is evidently challenging enough to produce an adverse mental health status. However, when it comes to resilience varying between male and female, this result could also inform us that the type of qualities that enhanced individual resilience didn’t differ according to the gender type. In other words, how male and female respondents answered the indicators within the resilience measure have no significant difference.
Reviewer 2 Report
There is a growing body of literature on the adjustment of refugees and asylum seekers. Using a unique sample of recent immigrants from the Republic of Korea (ROK), the present study adds to this stream of research by examining the cumulative impact of perceived discrimination, socio-economic strain, and structural strain on displaced people’s mental health. The authors also test the hypothesis that resilience mediates the relationship between post-migration life adversity and mental health problems. The sample includes 219 asylum seekers and 42 recognized refugees from 22 countries. The authors used structural equation modelling to assess the strength of the hypothesized pathways between the variables of interest. As expected, perceived discrimination, socio-economic strain, and structural strain showed a significant influence on mental health. Arguably, the most important finding in this study is resilience of refugees and asylum seekers exerts a mediating effect on the relationship between the post-migration life adversity on mental health problems. The authors conclude that fostering resilience among refugees and asylum seekers will likely improve their overall adjustment in the host country.
While being positive about the topic of this study, I must acknowledge that this study has several methodological drawbacks. First, it is not clear why and how the authors chose their sample. How many people were invited to participate? How many answered positively? What was the response rate? What were the criteria for participation? Where were the participants recruited from? Second, it is possible that some participants had mental problems before emigrating. Since the authors did not collect any information about the participants’ mental health in the origin countries, it is difficult to attribute mental health problems in the host countries solely to post-migration life adversity. Third, prior research shows that the effect of resilience on mental health largely depends on how resilience is measured. Resilience is a complex notion, and there is no agreement in the literature on how to operationalize this concept. The authors did not provide enough information concerning why they chose this particular scale to measure resilience. Fourth, they also provide little information concerning the theoretical basis for using perceived discrimination, socio-economic strain, and structural strain as predictors of mental health. Some of these predictors may be correlated. For example, structural strain can be affected by socio-economic strain. Additionally, past research shows that socio-economic strain may induce perceived discrimination.
Author Response
A response to reviewer 2
The research team for this paper wants to thank you for your interest and time to review our paper entitled ‘Post-migration life adversity and mental health of refugees and asylum seekers: The mediating role of resilience between perceived discrimination, socio-economic strains, structural strains, and mental health'. We appreciate all the comments given to the manuscript. They were highly insightful and very helpful in revising the study.
Here we will address your concerns one by one;
- How many people were invited to participate? How many answered positively? What was the response rate? What were the criteria for participation? Where were the participants recruited from?
A total of 350 (n=250 asylum seekers, n=100) survey invitation was sent out to potential participants. The eligible participants were selected by checking their immigrant status for the immigrant group and by checking national registration Korean status for Korean participants. A minimum of 6 month stay In South Korea was a mandatory participant selection criterion. Screening question for a past history of mental health problem before migrating to South Korea before this particular survey was asked and those with a history mental health were excluded from the survey.
- Your concern with history of mental health problem.
As mentioned in the answer to the first question, the screening question for a past history of mental health problem before migrating to South Korea was employed to avoid such bias.
- Third, prior research shows that the effect of resilience on mental health largely depends on how resilience is measured. Resilience is a complex notion, and there is no agreement in the literature on how to operationalize this concept. The authors did not provide enough information concerning why they chose this particular scale to measure resilience.
Our understanding of resilience for this particular study is the individual’s ability to effectively cope with significant adverse and threatening situations. The deduction of resilience in this study focuses on the individuals’ ability to “bounce back” to a previous state of normal functioning or using personal strengths and behaviors to avoid negative effects of adversities
The scale we used (AARS) addressed a major gap in previous literatures about how to measure the positive acculturation and resilience of the newly arrived and relocated individuals. The scale's psychometric properties designed for migrants and refugees from a range of ethnic communities are considered appropriate for our study to adopt it. And we included this rational within the revised manuscript.
- There is little information concerning the theoretical basis for using perceived discrimination, socio-economic strain, and structural strain as predictors of mental health.
We did not intentionally left out that theoretical elaboration. We however impliedly mentioned perceived discrimination, socio-economic strain, and structural strain as part of the post-migration experience (life adversity) of our study participants within South Korea. As a contextual background, our initial argument for possible factors for post-migration life adversities among refugees and asylum seekers emanate from the following ideal blocks. First, the post-migration experience of refugees and asylum seekers in South Korea, although different between the groups, is highly affected by the immigration policy, South Korean societal dynamics, and the overall refugee hosting mechanism (refugee status determination process and settlement programs). Second, due to such post-migration experience, mental health of refugees and asylum seekers is thus deemed to be shaped to a large extent by the mentioned social, economic and environmental factors. For better clarity, we also included this particular point alongside with the literature review.
Round 2
Reviewer 2 Report
The authors have attempted to answer all my questions and to incorporate all my suggestions to the best of their knowledge.